# The effect of psychological distress on IVF outcomes: Reality or speculations?

Gulzhanat Aimagambetova[1]*, Alpamys Issanov[2], Sanja Terzic[2], Gauri Bapayeva[3], Talshyn Ukybassova[3], Saltanat Baikoshkarova[4], Aidana Aldiyarova[3], Fariza Shauyen[3], Milan Terzic[2,3,5]

1 Department of Biomedical Sciences, Nazarbayev University School of Medicine, Nur-Sultan, Kazakhstan, 2 Department of Medicine, Nazarbayev University School of Medicine, Nur-Sultan, Kazakhstan, 3 Clinical Academic Department of Women's Health, National Research Center of Mother and Child Health, University Medical Center, Nur-Sultan, Kazakhstan, 4 IVF Clinic "Ecomed", Almaty, Kazakhstan, 5 Department of Obstetrics, Gynecology and Reproductive Sciences, University of Pittsburgh School of Medicine, Pittsburgh, Pennsylvania, United States of America

* gulzhanat.aimagambetova@nu.edu.kz

**Data Availability Statement:** The data are uploaded to Open Science Framework: osf.io/2bknw.

## Abstract

### Introduction

Infertility is a problem that affects millions of people worldwide. The aim of this study was to assess the effect of stress, depression and anxiety on the IVF outcomes in Kazakhstan.

### Methods

The prospective cohort study was performed using questionnaires to assess psychological distress in 304 infertile female in three different cities in Kazakhstan.

### Results

The average age of participants was 33.7 years with infertility duration of 5.9 years. Regarding stress, depression and anxiety we found that more than 80% of all respondents had CES-D score higher than 16, indicating that they are at risk of developing clinical depression. On average, FPI subscales' scores, global stress score and anxiety scale (STAI-S and STAI-T) scores were statistically significantly higher among not pregnant women than pregnant women. Similarly, in simple logistic regression analysis all FPI subscales scores, global stress scale score and anxiety scales' scores were negatively associated with clinical pregnancy.

### Conclusion

Rates of stress, anxiety and depression among IVF patients are higher than in general population. If the level of infertility-related stress is higher, IVF success rate is lower. Findings of our study indicate the need for the specific psychological interventions for all infertility women, to improve IVF success rate.

**Funding:** The research was supported by the Nazarbayev University Social policy grant awarded for author MT. Ecomed provided support for this study in the form of salary for author SB. The specific roles of these authors are articulated in the 'author contributions' section. The funders had no other role in study design, data collection and analysis, decision to publish, or preparation of the manuscript.

**Competing interests:** The authors have read the journal's policy and the authors of this manuscript have the following competing interests: SB is a paid employee of Ecomed. There are no patents, products in development or marketed products to declare. This does not alter our adherence to PLOS ONE policies on sharing data and materials.

## Introduction

Infertility is defined as failure to achieve pregnancy within twelve months of unprotected intercourse or therapeutic donor insemination in women younger than 35 years or within six months in women who are older than 35 years [1–3]. The problem affects about 9% of reproductive-age couples or more than 186 million people worldwide [2, 4], and up to 15% of couples in the USA [3].

Conception and reproduction is a foundation stone in most couples' lives. Thus, if a couple fails to conceive spontaneously, both partners' experiences sadness and disappointment. According to Sapolsky (2015), psychological status can activate the stress system response [5]. Infertility is one of the greatest factors in life that results in psychological stress. Anxiety, depression, and stress are the most frequently occurring psychological disorders among infertile patients [4, 6]. It has been hypothesized and confirmed that stress and stress-related factors have an influence on the autonomic, neuroendocrine and immune systems [4, 7–9]. Chronic stress has been found to induce sensitization of the hypothalamic-pituitary-adrenocortical (HPA) axis responsible for neuroendocrine functions [7], therefore potentially could have a negative impact on fertility, which in turn leads to the development of stress, anxiety and depression and the formation of a vicious circle. Supporting this statement, the length of time to conceive has been demonstrated to be associated with stress in infertile couples [4, 8, 10].

As it is clear now, infertility leads to stress, depression and anxiety. Many infertile couples go to seek In Vitro Fertilization (IVF) treatment. However, only a quarter of women will get pregnant after a single IVF cycle, so most couples will experience negative pregnancy results and repeat treatment. Rates of depression and anxiety increased after IVF treatment failure, while depression decreased after successful treatment [6, 11, 12]. However, both conditions—infertility and its treatment—cause stress, and it is well known that infertility can induce psychological disturbances [13]. The stress of infertility treatment was ranked second to that involving the death of a family member or divorce by couples undergoing this treatment [10, 13]. Moreover, patients who undergo assisted reproductive treatment (ART) are at high risk of developing psychiatric disorders. Thus, it is important to identify, acknowledge, and assist these patients as they cope with their infertility diagnosis and treatment [9].

Kazakhstan is one of the Central Asian republics which achieved independence after the Soviet Union was broken. The deep economic crisis of the early post-Soviet years was accompanied by a dramatic drop in fertility [14–16]. Population of Kazakhstan is multiethnic and composes about 19 million people with 26.7% of women in fertile age [17, 18]. Kazakhs, as a title ethnos, belong to the Turkic ethno-cultural group and have long-term patterns of family formation and fertility [14, 15]. The fertility cult in Central Asia is based on the desire to have many children (especially sons) and has had social and economic causes for thousands of years [16]. In the past, woman's fertility/infertility determined her status in the traditional society of Central Asia. Even today, women who have no children can be treated with scorn, resulting in loneliness and stigmatization from family or relatives site [16]. In the Republic of Kazakhstan, there is no reliable statistics on the frequency of infertile couples. According to various data, the frequency varies from 12 to 15.5% [17, 19, 20]. Both female and male infertility currently remains one of the most important challenges in reproductive endocrinology and infertility medicine in Kazakhstan [21, 22]. In spite of the prominent healthcare problems in this field, there is no any research done investigating the influence of the psychological distress on IVF treatment in Kazakhstan. Therefore, the aim of this study was to assess the effect of stress, depression and anxiety on the IVF outcomes in Kazakhstan.

## Materials and methods

### Study design

The prospective cohort study was performed from June 2019 to February 2020 in three IVF clinics in Kazakhstan with 2320 patients have been approached in total. Out of all approached patients 304 agreed to participate. These clinics were located in three large cities: Nur-Sultan (capital, population size ~1 mln), Almaty (former capital, population size ~ 2mln) and Shymkent (population size ~ 1 mln). Standardized clinical protocols were used, as all clinics were branches of one private medical organization. Women referred to initial or repeated IVF treatment at the clinics were provided with oral and written information about the study and asked to participate. Eligible participants were recruited from three fertility clinics in Kazakhstan, and met the following inclusion criteria: (1) they were seeking IVF, (2) were over 18 years old, (3) able to answer questions in Kazakh, Russian or English. Exclusion criteria: (1) not able to answer questions in Kazakh, Russian or English; (2) younger than 18 years old; (3) refuse to participate. This study was approved by the University Medical Center Institutional Research Ethics Committee and Nazarbayev University Institutional Research Ethics Committee. Written informed consent was obtained from each participant.

### Data collection

Outcome variable was defined as clinical pregnancy, a live intrauterine pregnancy identified by ultrasound scan at eight gestational weeks. The baseline questionnaire collected data about socio-demographic characteristics, such as, age, BMI, education level that was categorized according to International Standard Classification of Education (ISCED 4 –secondary high school, ISCED 5 –post-secondary non-tertiary education and ISCED 6 –bachelor/master level education). Also their past medical history information was collected, such as, comorbidities associated with infertility; infertility duration, which was defined as the time from the date of active child wish, or the date of last miscarriage to the date of the first IVF clinic visit; number of previous deliveries; number of previous miscarriages; number of intentional pregnancy interruptions and number of previous IVF cycles performed. The number of oocytes retrieved, the number of embryos transferred, as well as the cause of infertility (female (tubo-peritoneal, ovarian), male and mixed), the type of treatment protocol, fertilization and implantation rates were documented by a physician.

### Psychological status

The psychological status of the participants was defined in terms of depression, infertility stress and state and trait anxiety. Depression was measured using the Center for Epidemiological Studies Depression Scale (CES-D) developed by Radloff [23]. CES-D was chosen for the study as it has become a widely used clinical screening tool for the presence of depression [24]. CES-D is 20-item scale, where each item ranges between 0 and 3, with a maximum sum of score 60 indicating the highest level of depression. In addition, CES-D overall score was dichotomized using a cut-off 16 and above as at risk for clinical depression [25]. In this study, the Cronbach's alpha coefficient for CES-D scale was 0.94.

To measure levels of infertility stress, the Fertility Problem Inventory [26], a 46-item questionnaire, was used. FPI assesses five different aspects of infertility-related stress: social concerns, sexual concerns, relationship concerns, rejection of childfree lifestyle and need for parenthood. All of these FPI subscales contribute to cumulative global infertility stress score where a maximum sum of score could be as high as 276 indicating the highest level of infertility-related stress. In this study, the Cronbach's alpha coefficient for FPI global infertility stress

scale was 0.88 (social concern = 0.73, sexual concern = 0.72, relationship concern = 0.72, rejection of childfree lifestyle = 0.85 and need for parenthood = 0.77). Correlation analysis indicated that the subscales of social concern, sexual concern and relationship concern had moderate correlation coefficients (S1 Table).

Anxiety was assessed using Spielberger State-Trait Anxiety Inventory (STAI) [27], by 20 items, each ranging in score from 1 to 4. First 10 items (STAI State) measure state anxiety, whereas the last 10 items (STAI Trait) measure trait anxiety. Each anxiety state scales has a maximum sum score of 40, which indicates the highest anxiety level. In this study, the Cronbach's alpha coefficients for STAI-S and STAI-T subscales were 0.89 and 0.83, respectively.

All scales were translated in Russian and Kazakh languages by experienced researchers and then back translated to check appropriateness to the original versions. Correlation analysis showed that the scales were appropriately measuring depression, stress and anxiety without overlapping areas (S2 Table).

## Statistical analysis

In descriptive analysis, normally distributed numeric variables were summarized using means and standard deviations, while medians and ranges were additionally calculated for non-normally distributed numeric variables. Categorical variables were summarizes in frequencies and percentages. Bivariate analysis, testing relationships of the clinical pregnancy with continuous exposure variables, were performed utilizing independent Student t-test or Wilcoxon rank-sum test, where appropriate. Associations of the outcome variable with categorical exposure variables were examined by chi-square test or Fisher's exact test. The associations of CES-D, FPI and STAI scales with the clinical pregnancy were tested using simple and multiple logistic regression analysis. The women's BMI, education level, location, cause of infertility, comorbidity, infertility duration and number of previous IVF cycles were used to calculate adjusted effect in multiple regression models. Other covariates were not included to multicollinearity, quazi-complete separation (fertilization rate and implantation rate), statistically and clinically non-significance in the model building processes. To test intercorrelations between FPI, CES-D, STAI-S and STAI-T scales, Person's correlation coefficients were calculated. Finally, post hoc power analysis was performed to determine if insufficient sample size have played role in non-significant findings. Power was calculated based on alpha = 0.05, a mean difference or the minimally clinically important difference was approximated by one standard error of a measurement ("calculated as the standard deviation of the scale multiplied by the square root of one minus its reliability coefficient"). CES-D scale had the lowest power (32.6%); to obtain statistical power at the recommended 0.80 level sample size for CES-D scale would be needed to be 800. STAI-S (65.6%) and STAI-T (74.9%) had lower than the recommended statistical power level. All FPI subscales had sufficient power (>80%) except Need for parenthood (71.6%). All statistical analysis was done with STATA version 15 statistical software (StataCorp. 2017. Stata Statistical Software: Release 15. College Station, TX: StataCorp LLC). Statistical testing on the outcome variable was done at a 0.05 two-sided level of significance.

## Results

### Study sample

During the study 2320 patients have been approached in total in three IVF clinics. Out of all approached, 304 women who underwent IVF agreed to participate in the study. The average age of participants was 33.7 years old. The majority (64.0%) had normal BMI (18.5–24.9 kg/m2), education level at ISCED 5 or higher (77.3%, Table 1) and paid themselves (out-of-pocket) for IVF procedure (87.0%).

**Table 1. Demographic and socioeconomic characteristics of the study participants.**

| Variable | Total, N = 304 | Pregnant, n = 181 | Not pregnant, n = 48 | p-value |
|---|---|---|---|---|
| Age (years), mean±SD | 33.7±5.9 | 33.8±6.2 | 34.8±5.6 | 0.31 |
| Missing data = 1.3% | | | | |
| BMI, n(%) | | | | |
| - Underweight (less than 18.5 kg/m2) | 34 (12.9%) | 31 (17.8%) | 3 (6.4%) | 0.02 |
| - Normal (18.5–24.9 kg/m2) | 169 (64.0%) | 112 (64.4%) | 28 (59.6%) | |
| - Overweight/Obese (more than 25 kg/m2) | 61 (23.1%) | 31 (17.8%) | 16 (34.0%) | |
| Missing data = 13.2% | | | | |
| Education level, n(%) | | | | |
| ISCED 4 | 69 (22.7%) | 30 (16.6%) | 16 (33.3%) | 0.04 |
| ISCED 5 | 98 (32.2%) | 77 (42.5%) | 16 (33.3%) | |
| ISCED 6 | 137 (45.1%) | 74 (40.9%) | 16 (33.3%) | |
| Missing data = 0% | | | | |
| Location, n(%) | | | | |
| Almaty | 99 (32.6%) | 59 (60.8%) | 38 (39.2%) | <0.001 |
| Nur-Sultan | 108 (35.5%) | 103 (98.1%) | 2 (1.9%) | |
| Shymkent | 97 (31.9%) | 19 (70.4%) | 8 (29.6%) | |
| Missing data = 0% | | | | |

Approximately, one third of women had comorbidities and history of previous deliveries (Table 2).

Average infertility duration was 5.9 years. More than half infertility cases were attributed to female factor, and one fourth of women had previously attempted IVF cycles. The majority of women were treated using classic-short protocol (85.4%), (Table 3).

One woman had miscarriage (0.4%). The clinical pregnancy rate was almost 80% (25% missing data on clinical pregnancy), and 2% of pregnant women had multiple pregnancies as a result of ART.

## Comparing pregnant versus not pregnant women

Pregnant women were no different from not pregnant women in age (p = 0.31), income level (p = 0.78), having comorbidities (p = 0.43), infertility duration (p = 0.13), number of previous deliveries (p = 0.68), number of previous miscarriages (p = 0.51), number of previous intentional pregnancy interruptions (p = 0.94), number of previous IVF cycles (p = 0.15), treatment protocol (p = 0.57). However, the percentage of overweight/obese among pregnant women were twice lower than not pregnant (17.8% to 34.0%, respectively) whereas proportion of pregnant women with education level ISCED 5 and higher was statistically significantly more than not pregnant (83.4% to 66.6%, respectively). Women in Nur-Sultan city had the highest pregnancy rate than other cities (p<0.001). Number of oocytes retrieved (p = 0.05) and embryos transferred (p<0.01), as well as fertilization rate (0.04) and implantation rate (p<0.001) were statistically significantly associated with clinical pregnancy (Table 3).

## Regional differences

Even though the number of women selected from three cities were approximately equal, the distribution of their characteristics were not similar (Table 4).

More than one third of women from Shymkent city had education level at ISCED-5 (p<0.001), female factor as a cause of infertility were observed in four fifth of them (p<0.001)

**Table 2. Reproductive characteristics of the study participants.**

| Variable | Total, N = 304 | Pregnant, n = 181 | Not pregnant, n = 48 | p-value |
|---|---|---|---|---|
| Comorbidity, n(%) | | | | |
| Yes | 91 (29.9%) | 68 (37.6%) | 21 (43.8%) | 0.43 |
| No | 213 (70.1%) | 113 (62.4%) | 27 (56.2%) | |
| Missing data = 0% | | | | |
| Infertility duration (years) | | | | |
| Mean±SD | 5.9±4.1 | 5.2±3.8 | 6.3±4.4 | 0.13 |
| Median (IQR) | 5 (0–22) | 4 (0–22) | 5 (1–18) | |
| Missing data = 5.9% | | | | |
| Number of previous deliveries, n(%) | | | | |
| None | 192 (63.6%) | 114 (63.0%) | 31 (66.0%) | 0.68 |
| One | 76 (25.2%) | 45 (24.9%) | 9 (19.1%) | |
| Two or more | 34 (11.2%) | 22 (12.1%) | 7 (14.9%) | |
| Missing data = 0.7% | | | | |
| Number of previous miscarriages, n(%) | | | | |
| None | 257 (85.1%) | 157 (86.7%) | 39 (83.0%) | 0.51 |
| One | 45 (14.9%) | 24 (13.3%) | 8 (17.0%) | |
| Missing data = 0% | | | | |
| Number of previous intentional pregnancy interruptions, n(%) | | | | |
| None | 279 (92.4%) | 165 (91.2%) | 43 (91.5%) | 0.94 |
| One | 16 (7.6%) | 16 (8.8%) | 4 (8.5%) | |
| Missing data = 0.7% | | | | |
| Number of previous IVF cycles, n(%) | | | | |
| None | 229 (76.1%) | 149 (82.8%) | 34 (70.8%) | 0.15 |
| One | 42 (13.9%) | 16 (8.9%) | 6 (12.5%) | |
| Two or more | 30 (10.0%) | 15 (8.3%) | 8 (16.7%) | |
| Missing data = 1.0% | | | | |
| Cause of infertility, n(%) | | | | |
| Female | 161 (53.5%) | 78 (43.3%) | 26 (54.2%) | <0.01 |
| Male | 33 (11.0%) | 14 (7.8%) | 11 (22.9%) | |
| Mixed | 107 (35.5%) | 88 (48.9%) | 11 (22.9%) | |
| Missing data = 1.0% | | | | |

while the average number of embryos transferred was the highest for Shymkent city among other cities (p<0.001). On the other hand, the percentage of overweight or obese women were not as high as in Almaty city in relation to Shymkent city (34.7% versus 25.4%, respectively, p<0.001).

## Depression, stress and anxiety

More than 80% of all respondents had CES-D score higher than 16, indicating that they are at risk of developing clinical depression (Table 4). Since there are no available Kazakhstani country or regional based data on the CES-D scores, the calculations were done based on the authors' recommendations [25]. Depression scale score was not different between pregnant and not pregnant women (25.1 to 26.7, respectively, p = 0.26), (Table 5).

On average, women from Almaty city had the highest depression scale score (p<0.001), (Table 6).

**Table 3. IVF treatment characteristics of the study participants.**

| Variable | Total, N = 304 | Pregnant, n = 181 | Not pregnant, n = 48 | p-value |
|---|---|---|---|---|
| Number of oocytes retrieved | | | | |
| Mean±SD | 11.5±8.4 | 11.7±8.2 | 9.5±7.8 | 0.05 |
| Median (IQR) | 10 (0–49) | 10 (1–49) | 7 (0–30) | |
| Missing data = 7.9% | | | | |
| Number of embryos transferred | | | | |
| Mean±SD | 2.2±2.5 | 1.6±0.7 | 2.3±2.6 | <0.01 |
| Median (IQR) | 2 (0–18) | 2 (1–7) | 2 (0–18) | |
| Missing data = 15.1% | | | | |
| Used protocol | | | | |
| Classic-long | 31 (10.3%) | 21 (11.6%) | 9 (18.7%) | 0.57 |
| Classic-short | 257 (85.4%) | 149 (82.3%) | 37 (77.1%) | |
| Non-classic—natural cycle | 5 (1.6%) | 4 (2.2%) | 1 (2.1%) | |
| Non-classic—ultrashort | 8 (2.7%) | 7 (3.9%) | 1 (2.1%) | |
| Missing data = 1.0% | | | | |
| Fertilization rate, % | | | | |
| Mean±SD | 85.6±10.2 | 86.0±10 | 84.0±20 | 0.04 |
| Median (IQR) | 90 (0–98) | 90 (60–98) | 90 (0–90) | |
| Missing data = 43.8% | | | | |
| Implantation rate, % | | | | |
| Mean±SD | 68.6±19.4 | 67±20 | 84±20 | <0.001 |
| Median (IQR) | 60 (0–98) | 60 (30–98) | 91 (0–96) | |
| Missing data = 43.8% | | | | |

**Table 4. Demographic and socioeconomic characteristics of the study participants by location.**

| Variable | Almaty, n = 99 | Nur-Sultan, n = 108 | Shymkent, n = 97 | p-value |
|---|---|---|---|---|
| BMI, n(%) | | | | |
| Underweight | 10 (10.2%) | 24 (22.4%) | 0 (0.0%) | <0.001 |
| Normal | 54 (55.1%) | 71 (66.4%) | 44 (74.6%) | |
| Overweight/Obese | 34 (34.7%) | 12 (11.2%) | 15 (25.4%) | |
| Education level, n(%) | | | | |
| ISCED 4 | 15 (15.1%) | 20 (18.5%) | 34 (35.1%) | <0.001 |
| ISCED 5 | 46 (46.5%) | 45 (41.7%) | 7 (7.2%) | |
| ISCED 6 | 38 (38.4%) | 43 (39.8%) | 56 (57.7%) | |
| Cause of infertility, n(%) | | | | |
| Female | 45 (45.4%) | 39 (36.8%) | 77 (80.2%) | <0.001 |
| Male | 15 (15.1%) | 5 (4.7%) | 13 (13.5%) | |
| Mixed | 39 (39.4%) | 62 (58.5%) | 6 (6.3%) | |
| Number of oocytes retrieved | | | | |
| Mean±SD | 11.4±8.8 | 11.8±7.8 | 11.2±8.8 | 0.48 |
| Median (IQR) | 9 (1–40) | 10 (0–49) | 9 (1–43) | |
| Number of embryos transferred | | | | |
| Mean±SD | 1.7±0.7 | 1.6±0.5 | 4.5±4.8 | <0.001 |
| Median (IQR) | 2 (0–3) | 2 (0–2) | 2 (1–18) | |

**Table 5. Differences between pregnant and non-pregnant women on depression, stress and anxiety.**

| Scales | Total, N = 304 | Pregnant, n = 181 | Not pregnant, n = 48 | p-value |
|---|---|---|---|---|
| CES-D score, mean±SD | 25.2±10.0 | 25.1±12.1 | 26.7±12.1 | 0.37 |
| Categorized CES-D score, n (%) | | | | |
| No risk for clinical depression (< 16) | 48 (17.4%) | 38 (21.2%) | 9 (19.6%) | 0.80 |
| At risk for clinical depression ($\geq$ 16) | 228 (82.6%) | 141 (78.8%) | 37 (80.4%) | |
| Missing data = 9.2% | | | | |
| FPI scale | | | | |
| Social concern, mean±SD | 33.3±5.6 | 32.7±5.8 | 36.4±4.2 | <0.001 |
| Sexual concern, mean±SD | 25.7±6.0 | 24.5±6.1 | 29.9±4.1 | <0.001 |
| Relationship concern, mean±SD | 32.7±7.2 | 31.7±7.8 | 36.4±3.9 | <0.001 |
| Need for parenthood, mean±SD | 41.7±7.3 | 41.1±7.6 | 45.2±7.1 | <0.01 |
| Rejection of childfree lifestyle, mean±SD | 31.5±8.0 | 30.9±7.9 | 37.8±8.5 | <0.001 |
| Global stress, mean±SD | 164.9±21.8 | 160.9±21.9 | 185.7±17.0 | <0.001 |
| Missing data = 10.5% | | | | |
| STAI State, mean±SD | 47.1±9.7 | 45.1±9.9 | 49.9±7.2 | <0.01 |
| STAI Trait, mean±SD | 48.8±7.5 | 47.9±7.8 | 50.9±7.1 | 0.02 |
| Missing data = 13.2% | | | | |

No statistically significant association was found between depression scale score and clinical pregnancy outcome in simple and multiple logistic regression analysis (Table 7).

On average, women from Almaty city had the highest FPI stress subscales' scores and global stress score (p<0.001), (Table 6). In bivariable analysis, FPI subscales' scores and global stress score were statistically significantly higher among not pregnant women than pregnant women (Table 5). Similarly, in simple logistic regression analysis all FPI subscales scores and global stress scale score were negatively associated with clinical pregnancy outcome (Table 7). After adjusting for BMI, location, education level, cause of infertility, comorbidity, infertility duration and number of previous IVF cycles, only the higher stress related to sexual concern (p = 0.03), need for parenthood (p = 0.01), rejection of childfree lifestyle (marginally

**Table 6. Differences on depression, stress and anxiety between cities.**

| Scales | Almaty | Nur-Sultan | Shymkent | p-value |
|---|---|---|---|---|
| CES-D score, mean±SD | 28.7±11.9 [ns, sh] | 22.9±9.5 [al, sh] | 23.8±5.4 [ns, al] | <0.001 |
| Categorized CES-D score, n (%) | | | | |
| At risk for clinical depression ($\geq$ 16) | 82 (82.8%) | 80 (74.1%) | 66 (95.7%) | <0.01 |
| FPI scale | | | | |
| Social concern, mean±SD | 35.9±3.6 [ns, sh] | 31.1±6.4 [al, sh] | 33.2±4.9 [ns, al] | <0.001 |
| Sexual concern, mean±SD | 29.2±4.1 [ns, sh] | 21.6±6.0 [al, sh] | 26.9±4.7 [ns, al] | <0.001 |
| Relationship concern, mean±SD | 36.3±4.1 [ns, sh] | 29.0±8.7 [al, sh] | 33.4±4.9 [ns, al] | <0.001 |
| Need for parenthood, mean±SD | 44.1±6.9 [ns, sh] | 40.5±8.0 [al] | 40.2±5.9 [al] | <0.001 |
| Rejection of childfree lifestyle, mean±SD | 36.9±9.0 [ns, sh] | 29.2±6.6 [al] | 27.8±3.8 [al] | <0.001 |
| Global stress, mean±SD | 182.3±16.4 [ns, sh] | 151.4±20.5 [al, sh] | 161.5±12.5 [ns, al] | <0.001 |
| STAI State, mean±SD | 46.9±8.6 [ns, sh] | 43.1±10.0 [al, sh] | 54.1±6.0 [ns, al] | <0.001 |
| STAI Trait, mean±SD | 49.7±7.0 [ns] | 46.4±8.2 [al, sh] | 51.4±5.7 [ns] | <0.001 |

[ns]—Average scale score was statistically significantly different from Nur-Sultan city.

[al]—Average scale score was statistically significantly different from Almaty city.

[sh]—Average scale score was statistically significantly different from Shymkent city.

**Table 7. Simple and multiple logistic regression analyses of psychological factors predicting IVF clinical pregnancy.**

| Scales | OR$_{Crude}$ (95% CI) | p-value | *OR$_{Adj}$ (95% CI) | p-value |
|---|---|---|---|---|
| CES-D score | 0.99 (0.96–1.02) | 0.37 | 1.02 (0.98–1.07) | 0.35 |
| FPI scale | | | | |
| Social concern | 0.87 (0.80–0.94) | <0.001 | 0.92 (0.82–1.03) | 0.17 |
| Sexual concern | 0.83 (0.77–0.89) | <0.001 | 0.88 (0.79–0.99) | 0.03 |
| Relationship concern | 0.90 (0.85–0.95) | <0.001 | 0.92 (0.84–1.01) | 0.10 |
| Need for parenthood | 0.93 (0.89–0.97) | <0.01 | 0.91 (0.85–0.98) | 0.01 |
| Rejection of childfree lifestyle | 0.90 (0.86–0.94) | <0.001 | 0.95 (0.90–1.00) | 0.05 |
| Global stress | 0.94 (0.93–0.96) | <0.001 | 0.95 (0.93–0.98) | <0.01 |
| STAI State | 0.94 (0.91–0.98) | <0.01 | 0.97 (0.91–1.02) | 0.22 |
| STAI Trait | 0.95 (0.91–0.99) | 0.02 | 0.99 (0.93–1.05) | 0.66 |

* Each of the scales was adjusted for BMI, location, education level, cause of infertility, comorbidity, infertility duration and number of previous IVF cycles.

associated, p = 0.05) and global stress (p<0.01) women experienced were statistically significantly negatively associated with clinical pregnancy outcome (Table 7).

Shymkent city respondents had the highest STAI anxiety scales' scores (p<0.001) than other respondents. On average, anxiety scale (STAI-S and STAI-T) scores were statistically significantly higher among not pregnant women than pregnant women. In simple logistic regression analysis, anxiety scales' scores were negatively associated with clinical pregnancy outcome, however, no associations were detected in multiple logistic regression analysis (Table 7).

## Discussion

This is the first multicenter study in Kazakhstan assessing correlation of infertility-related stress, anxiety and depression with the IVF outcome. Despite apparent differences in distribution of factors associated with the IVF outcome among the cities, baseline stress was independently associated with the clinical pregnancy. This study indicates that the lower clinical pregnancy rate was negatively associated with sexual concern, need for parenthood and global infertility stress. These results are in line with previous research. Some authors [28] found that stress was associated with reduced fertilization, implantation and live birth rates. Notably, in our study, infertility-related stress levels were higher in south regions (Almaty and Shymkent), where fertility and children are highly valued. In South Kazakhstan, women are likely blamed for infertility, and motherhood is the only way women can gain status within a family and community. Until then, women are stigmatized for inability to conceive, and they are at higher risk of domestic violence and disrespectful treatment by husband and relatives [29].

Previous studies have showed that release of stress hormones were negatively association with IVF treatment outcomes [28, 30, 31]. Also, there is some evidence that stress is associated with reduced fertilization, implantation and live birth rates [32, 33]. Our study did not find statistically significant association of anxiety and depression with the pregnancy rate of IVF. Even though in bivariate analysis state and trait anxiety were associated with unsuccessful IVF outcomes, after adjusting for other independent variables the associations became insignificant. In this regard, results from previous studies are inconsistent. A systematic review found weak negative association of pregnancy rates with anxiety and no association with depression [11]. However, the authors pointed out in limitations of the included studies, such as, various study designs, different tools used to measure psychological status, differences in inclusion criteria,

and the majority of the studies had failed to control for important factors. It is still unclear whether these associations are present, thus, future well-designed studies are needed.

Even though no association of anxiety and depression with IVF outcome was found, high rates of anxiety and being at risk of developing clinical depression were observed among all study participants, regardless of their IVF outcomes. More than 80% of women indicated of having moderate to severe depressive symptoms. Also, state and trait anxiety levels in our study were statistically significantly higher than normative population. Similarly, other studies have found high rates of depression and anxiety among subfertile women before IVF treatment [34–37].

In this study, women who failed to conceive after IVF treatment more likely were overweight or obese than those who were successful. Similar results are obtained from other studies [38–40]. Overweight or obese women are at higher risk of IVF failures attributing to menstrual dysfunction, hormonal imbalance [39] or lower incidence of embryo implantation [38]. A systematic review found that non-overweight women had 40% higher odds of pregnancy rate than overweight women [40]. Also, we found that higher education level is associated with better IVF outcome. It is believed that higher education level is related to other contributing factors: healthy life behaviors [41], higher income [42] and better compliance to treatment regime [43].

Our study results showed that clinical IVF outcomes were interrelated with each other. Number of oocytes retrieved, number embryos transferred, fertilization rate and implantation rate were associated with clinical pregnancy. However, some IVF outcomes were reversely associated with each other. Given large proportion of data was missing for implantation rate, observed unexpected negative association with IVF outcome and positive correlation with stress should be considered with caution.

## Strengths and limitations

This is the first study in the Central Asia region examining possible associations of the IVF outcome with infertility stress, anxiety and depression before commencing IVF treatment, taking into account other confounding factors that independently associated the IVF outcome. This study includes sufficiently large statistical power to detect differences in infertility stress scale between pregnant and non-pregnant women. Our study used internationally validated questionnaires to determine baseline psychological status. Scales had sufficient to high reliability scores.

Nonetheless, this study is not without limitations. It was not possible to investigate relationships of psychological status with other clinical IVF outcomes, such as, fertilization rate and implantation rate given insufficient data for analysis. It could be interesting to examine whether infertility stress, anxiety and depression were independently associated with fertilization rate and implantation rate, controlling for IVF treatment type and medications dosage. Non-response bias could be another limitation, as data on non-respondents were not collected for comparison. It is unknown whether non-respondents were different from those who participated in the study in terms of psychological status and IVF outcomes. For example, participants from Nur-Sultan city had the most successful pregnancy rate (98.1%) while psychological status scores were lower than other cities. Other factors were also not collected, such as, alcohol and caffeine consumption and smoking, sport activities practicing, type of profession and the length of sedentary period during the day, environmental and working place' factors and other factors. It is understood that these lifestyle, working and environmental factors could be mediating the effects of stress, anxiety and depression of the IVF outcome and should be analyzed differently. Low statistical power because of small sample size in the current study may have played a role in limiting finding the significant difference in

depression between pregnant and non-pregnant women. Lastly, due to a relatively small sample size it was not possible to investigate the relationships of stress, anxiety and depression with IVF outcome stratifying by first and repeated IVF cycle groups (as well as to assess the stress, anxiety and depression in patients coming for the repeated IVF cycle with previous unsuccessful performed in the actual clinic, in other private clinical setting or in the public clinical setting). Women with repeated IVF cycles, unlike first-time IVF patients, have previous psychological experience of IVF failures and are likely to have unfavorable IVF outcomes. Inclusion of repeat IVF patients could potentially pull the association away from null, which in turn overestimate the true relationship.

## Conclusions

Our results illustrate that infertility-related stress is associated with the clinical pregnancy, as a successful IVF outcome. However, no clear relationship was found with anxiety or depression. Additionally, IVF patients in the south region experience a higher level of infertility-related stress, anxiety, and depression (where reproduction expectations were not fulfilled and the infertility couple experiences the persistent pressure from the family and relatives). And overall, rates of depression and anxiety among IVF patients are higher than general population. Findings of our study point out the necessity of specific psychological interventions for all subfertile women, especially providing psychological support in south regions. These interventions will improve of mental health and will help to achieve maternal goals.

## Supporting information

**S1 Table. Intercorrelations between Fertility Problem Inventory (FPI) subscales.**
(DOCX)

**S2 Table. Correlations of Fertility Problem Inventory (FPI) scale with depression (CES-D Scale), state anxiety (STAI-S) and trait anxiety (STAI-T) scales.**
(DOCX)

## Acknowledgments

The authors acknowledge the Nazarbayev University School of Medicine for creating supportive working atmosphere that enabled the completion of this research.

## Author Contributions

**Conceptualization:** Gauri Bapayeva, Milan Terzic.

**Data curation:** Gulzhanat Aimagambetova, Sanja Terzic, Talshyn Ukybassova, Saltanat Baikoshkarova, Aidana Aldiyarova, Fariza Shauyen.

**Formal analysis:** Gulzhanat Aimagambetova, Alpamys Issanov, Aidana Aldiyarova.

**Funding acquisition:** Milan Terzic.

**Investigation:** Gulzhanat Aimagambetova, Sanja Terzic, Talshyn Ukybassova, Saltanat Baikoshkarova, Fariza Shauyen.

**Methodology:** Alpamys Issanov, Milan Terzic.

**Project administration:** Talshyn Ukybassova, Milan Terzic.

**Resources:** Milan Terzic.

**Software:** Alpamys Issanov.

**Supervision:** Gulzhanat Aimagambetova, Gauri Bapayeva, Talshyn Ukybassova, Saltanat Baikoshkarova, Milan Terzic.

**Validation:** Gulzhanat Aimagambetova, Alpamys Issanov.

**Visualization:** Aidana Aldiyarova, Fariza Shauyen.

**Writing – original draft:** Gulzhanat Aimagambetova, Alpamys Issanov, Sanja Terzic.

**Writing – review & editing:** Gulzhanat Aimagambetova, Milan Terzic.

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
