## [Decision Letter · Decision Letter 0]

1 Sep 2020

PONE-D-20-18832

The effect of psychological distress on IVF outcomes: reality or speculations?

PLOS ONE

Dear Dr. Aimagambetova,

Thank you for submitting your manuscript to PLOS ONE. After careful consideration, we feel that it has merit but does not fully meet PLOS ONE’s publication criteria as it currently stands. Therefore, we invite you to submit a revised version of the manuscript that addresses the points raised during the review process.

Both reviewers highlighted interesting points that need to be addresses by the Authors. In particular, points 1 and 3 raised by Reviewer 2 deserve great consideration. In addition, a sample size calculation is needed in order to verify whether the study has an appropriate power to support the conclusions.

We look forward to receiving your revised manuscript.

Kind regards,

Alessio Paffoni, PhD

Academic Editor

PLOS ONE

Journal Requirements:

a) Did participants provide their written or verbal informed consent to participate in this study?

"No authors have competing interest"

We note that one or more of the authors are employed by a commercial company: IVF Clinic “Ecomed”.

3.1. Please provide an amended Funding Statement declaring this commercial affiliation, as well as a statement regarding the Role of Funders in your study. If the funding organization did not play a role in the study design, data collection and analysis, decision to publish, or preparation of the manuscript and only provided financial support in the form of authors' salaries and/or research materials, please review your statements relating to the author contributions, and ensure you have specifically and accurately indicated the role(s) that these authors had in your study. You can update author roles in the Author Contributions section of the online submission form.

3.2. Please also provide an updated Competing Interests Statement declaring this commercial affiliation along with any other relevant declarations relating to employment, consultancy, patents, products in development, or marketed products, etc.  

4. We note you have included a table to which you do not refer in the text of your manuscript. Please ensure that you refer to Table 5 and 6 in your text; if accepted, production will need this reference to link the reader to the Tables.

Reviewers' comments:

Reviewer's Responses to Questions

**Comments to the Author**

1. Is the manuscript technically sound, and do the data support the conclusions?

Reviewer #1: Partly

Reviewer #2: Yes

2. Has the statistical analysis been performed appropriately and rigorously? 

Reviewer #1: I Don't Know

Reviewer #2: Yes

3. Have the authors made all data underlying the findings in their manuscript fully available?

Reviewer #1: Yes

Reviewer #2: Yes

4. Is the manuscript presented in an intelligible fashion and written in standard English?

Reviewer #1: No

Reviewer #2: Yes

5. Review Comments to the Author

Reviewer #1: 1. Because patient distress can be highly correlated to their perception of their prognosis, the analysis should include a control for prognosis, such as AMH, AFC, FSH, etc

2. How many women were approached to get the total of 304?

3. I find the pregnancy rates to be exceptionally high. An average of 80% is extremely unusual. Is this the normal average pregnancy rate in the clinic?

4. please use a more updated reference for reference 29. Pasch et al have published more recent data in F&S

The authors repeatedly state that stress, anxiety, and depression were correlated with pregnancy but depression was not.

5. The pregnancy rate of 08.1% in one of the subgroups feels unreal. Especially when an average of 2 embryos were transferred. Please reassess these results.

6. BMI should be controlled for in the analysis.

Reviewer #2: Review comments

Title: The effect of psychological distress on IVF outcomes: reality or speculations?

This study investigates the rates of depression, anxiety and stress in women undergoing fertility treatement in Kazakhstan. Furthermore the present study explored if increased levels of depressive symptoms and anxiety influenced pregnancy outcome. Previously there haven't been exact numbers of how many women are burdened by the infertility or treatment nor if it has had any negative influence on pregnancy rates for women in the region of Kazakhstan, so it is very admirable that the authors seek to investigate. However there are some structural and language issues that need attention.

General comments: There are places in the manuscript where the argumentation is very sparse. It would be better if you leave out some of the points and build the understanding for the points left rather than a lot of points with no reasoning. Then it becomes difficult to see the relevance of the point. It could be interesting if the authors disclosed a little more of the importance of why infertility might be so devastating for women in the region of central asia - some information is discussed in the end of the manuscript. I find this information very important and interesting. Also, discussing admissability to treatment.

Introduction

L. 72-74: It would be good if the authors explained from where they got these definitions of infertility, although they put in a reference, as there are differenct definitions according to country and region.

L. 79-80: I would advise the authors to discuss the terms stress and distress, and when to use the nomination. It seems that due to sadness and dissapointment one would experience distress (emotional burden), whereas treatment in itself can be stressing and probably lead to distress.

L. 81-84: The aregumenation is a little too superficial. I agree that stress and distress may influence autominic and endocrine responses, and this, is defintely very interesting. However, the argumentation lacks examples or referral to the specific studies - the 'how'.

L. 85-88: This is true - but this paragraph seem to be associated with former paragraphs explaining that couples are distressed or that infertility and fertility treatment might be stressing. Be aware that you don't jump around with different themes.

L.103: What does REI stand for?

Methods

L- 112: How many patients were approached? Did the women themselves decide if they wanted to take part in teh study?

L. 118: You might want to specify that two clinics were in the same city??? 3 clinics - two cities.

L. 125: When did they fill-out the baseine questionnaire?

L. 129: What kind of medical history information was gathered? Previous depressions, anxiety - treatments? Prescribed medication?

L.140-146: Why did you choose CES-D? And what is the average score of the Kazakhstan population? or regional data?

L.187: I believe that The Results section should be a header in itself and not presented under 'methods'.

Results

L. 189: How many did you approach - how many declined to particioate? Do you know why? A flowchart will give us a nice overview.

L. 190: definition of normal BMI (also in a cultural perspective). Also it is unclear, if women who conceived were in first, second or more cycles. It would be beneficial to add this information to your table (the frequency of pregnacy rate for each cycle).

L. 220-222: from what perspective? Do you have any regional data to compare with? Again a table with an overview of the descriptive reginal data would make it much easier to understand the differences.

L.230-233: I am a little usure that I understand this paragraph correctly. If you controlled for the variables mentionned then the association between FPI global stress was still significant or not significant for pregnancy outcome? You may want to consider 'main' analyses and then explorative analyses e.g. looking into specific domains association with pregnancy. It is a little muddled.

Discussion

Overall, the discussion is long. You may want to shorten it a bit.

L.251-261: This is a very interesting discussion point indeed, however as you don't yourselves present any biological/physiological data it seems a little odd to present it here. You are speculating whether this might be an explanation for your findings.

L.272-281: What is the main reason for this paragraph? You might want to leave it out. As it doesn't add to your main objective.

L. 301: If you did power calculations accordingly to the number of participants you needed, it would be a good idea to present this information in the Methods section, as this is defintely a strength.

L. 313-315: This is an interesting result, wy isn't this dicussed further?

6. PLOS authors have the option to publish the peer review history of their article (what does this mean?). If published, this will include your full peer review and any attached files.

Reviewer #1: No

Reviewer #2: No

---

## [Author Response · Author response to Decision Letter 0]

16 Oct 2020

Reviewer #1:

Dear Reviewer,

Thank you very much for the detailed review of our manuscript. We appreciate a lot your valuable comments and suggestions. Please find below our point by point responses for all your comments.

1. Because patient distress can be highly correlated to their perception of their prognosis, the analysis should include a control for prognosis, such as AMH, AFC, FSH, etc

Response: The Reviewer has raised the important issue: relationship between the level of hormones before the procedure and the patient distress (patients’ “perception of their prognosis”). Such analysis is a complex one, considering the interrelationship between hormonal levels and: 1. Number of oocytes retrieved, 2. Number of embryos transferred, 3. Protocol used, 4. Fertilization rate, 5. Implantation rate, 6. BMI, 7. Age, and other very im-portant data from the patient’s record. Also, considering that medications used for IVF ap-proach cause selective desensitization of GnRH cells of adenohypophysis, it might be also assumed that hormonal levels might not be related to IVF outcomes in all patients. Furthermore, hormonal status might be correlated with distress (stress, anxiety and depression) measured before the IVF procedure, immediately after the procedure, and finally when the result of beta-hCG has been attained and given to the patient. “The aim of this study was to assess the effect of stress, depression and anxiety on the IVF outcomes in Kazakhstan”, while all these mentioned very complex research assessments and interpretation might be eventually performed in the future, for PhD thesis in this field.

2. How many women were approached to get the total of 304?

Response: ECOMED is the first and the most developed IVF clinic in Kazakhstan with the representing branches in almost all regions of the country. More than 12000 newborns were delivered with the facilitation of ECOMED. The estimated number of patients per month is 1250.

To achieve 304 patients involved in the project we have approached 2¬320 patients in three branches of ECOMED clinic.

3. I find the pregnancy rates to be exceptionally high. An average of 80% is extremely un-usual. Is this the normal average pregnancy rate in the clinic? 

Response:

Out of 304 women, who attended IVF clinics, the clinical outcome was not determined for 75 (25%) of them, whereas the rest of the 229 study participants had pregnancy rate approximately 80%. It is possible that the pregnancy rate for the study participants could be overestimated as we do not know clinical outcomes for those 25% missing.

4. Please use a more updated reference for reference 29. Pasch et al have published more recent data in F&S. 

Response:

More recent citations have been included. 

Pasch LA, Sullivan KT. Stress and coping in couples facing infertility. Curr Opin Psychol. 2017;13:131-135. doi:10.1016/j.copsyc.2016.07.004

Pasch LA. New realities for the practice of egg donation: a family-building perspective. Fertil Steril. 2018;110(7):1194-1202. doi:10.1016/j.fertnstert.2018.08.055

5. The pregnancy rate of 08.1% in one of the subgroups feels unreal. Especially when an average of 2 embryos were transferred. Please reassess these results.

Response:

We thank the Reviewer for the comment. The results were reassessed. Indeed, 98.1% pregnancy rate for Nur-Sultan city was correctly calculated. Attached below the results:

This high rate could be explained by possible pre-selection of women with better future outcomes for IVF procedure in this city. We acknowledged that this selection bias by location could confound the relationship of psychological status and clinical pregnancy, thus, we included location in the multivariable logistic regression analysis to obtain less biased estimates.

6. BMI should be controlled for in the analysis.

Response: BMI was reported for all patients, presented in Table 1, and discussed in the first paragraph of the Results section.

In multivariable logistic regression analysis, the models were adjusted for BMI, location, education level, cause of infertility, comorbidity, infertility duration and number of previous IVF cycles. This statement was previously indicated in the footnote of Table 7. Also it was stated in Methods section “The women’s BMI, education level, location, cause of infertility, comorbidity, infertility duration and number of previous IVF cycles were used to calculate adjusted effect in multiple regression models.”

Reviewer #2:

Dear Reviewer,

Thank you very much for the detailed review of our manuscript. We appreciate a lot your valuable comments and suggestions. Please find below our point by point responses for all your comments.

Title: The effect of psychological distress on IVF outcomes: reality or speculations?

This study investigates the rates of depression, anxiety and stress in women undergoing fertility treatement in Kazakhstan. Furthermore the present study explored if increased levels of depressive symptoms and anxiety influenced pregnancy outcome. Previously there haven't been exact numbers of how many women are burdened by the infertility or treatment nor if it has had any negative influence on pregnancy rates for women in the region of Kazakhstan, so it is very admirable that the authors seek to investigate. However there are some structural and language issues that need attention.

General comments: There are places in the manuscript where the argumentation is very sparse. It would be better if you leave out some of the points and build the understanding for the points left rather than a lot of points with no reasoning. Then it becomes difficult to see the relevance of the point. It could be interesting if the authors disclosed a little more of the importance of why infertility might be so devastating for women in the region of central asia - some information is discussed in the end of the manuscript. I find this infor-mation very important and interesting. 

Response: Some information covering culture of fertility in Central Asia and Kazakhstan has been included into the text as was suggested by the Reviewer. 

“Kazakhstan is one of the Central Asian republics which achieved independence after the Soviet Union was broken. The deep economic crisis of the early post-Soviet years was accompanied by a dramatic drop in fertility [14,15]. Population of Kazakhstan is multiethnic and composes about 19 million people with 26.7% of population of women in fertile age [17,18]. Kazakhs, as a title ethnic group, belong to the Turkic ethno-cultural group and have long-term patterns of family formation and fertility [14,15]. The fertility cult in Central Asia is based on the desire to have many children (especially sons) has had social and economic causes for thousands of years [16]. In the past, woman’s fertility/infertility determined her status in the traditional society of Central Asia. Even today, women who have no children can be treated with scorn, resulting in loneliness and stigmati-zation from family or relatives site [16].”

Introduction

L. 72-74: It would be good if the authors explained from where they got these definitions of infertility, although they put in a reference, as there are differenct definitions according to country and region.

Response: There is no specific definition for infertility in the country or region. With re-gards to infertility, local medicine/OBGYN specialists are guided by international defini-tions/standards.

The definition has been obtained from doi:10.1097/AOG.0000000000003271 sited in the references [3]. Similar definitions are present in the references [1] and [2].

L. 79-80: I would advise the authors to discuss the terms stress and distress, and when to use the nomination. It seems that due to sadness and dissapointment one would experi-ence distress (emotional burden), whereas treatment in itself can be stressing and proba-bly lead to distress.

Response:

Stress was defined many decades ago by the endocrinologist Hans Selye as being related to the physiological adaptive response to perceived (psychological) or real (physical) threats (stressors) to an organism trying to re-establish the homeostatic dynamic equilibrium challenged by such threats.

According to Sapolsky (2015), the stress system response can be mainly activated by psychological states, including loss of control, predictability and social support, often increasing the probability of driving an organism to sickness.

Generally, as mentioned above, two main variants of stress are present, namely psychological and physical stress. In the manuscript we discuss psycological stress.

The term stress in discussed in the updated text.

L. 81-84: The aregumenation is a little too superficial. I agree that stress and distress may influence autominic and endocrine responses, and this, is defintely very interesting. However, the argumentation lacks examples or referral to the specific studies - the 'how'.

Response: More argument based on the references have been included to support the idea.

“Infertility is one of the greatest factors in life that results in psychological stress. Anxiety, depression, and stress are the most frequently occurring psychological disorders among infertile patients [4,6]. It has been hypothesized and confirmed that stress and stress related factors have an influence on the autonomic, neuroendocrine and immune systems [4,7-9]. Chronic stress has been found to induce sensitization of the hypothalamic-pituitary-adrenocortical (HPA) axis responsible for neuroendocrine functions [7], therefore potentially could have a negative impact on fertility, which in turn leads to the development of stress, anxiety and depression and the formation of a vicious circle.”

L. 85-88: This is true - but this paragraph seem to be associated with former paragraphs explaining that couples are distressed or that infertility and fertility treatment might be stressing. Be aware that you don't jump around with different themes.

Response: We agree with the Reviewer’s comment. The paragraph mentioned by the Reviewer has been deleted as a redundant.

L.103: What does REI stand for?

Response: REI medicine – reproductive endocrinology and infertility medicine. Amend-ments have been done in the text.

Methods

L- 112: How many patients were approached? Did the women themselves decide if they wanted to take part in teh study?

Response:

During the study 2320 patients have been approached in total in three IVF clinics. Out of all approached patients 304 agreed to participate. 

The decision to participate or not was made by women after the careful explanation of the study purposes. The decision to participate was absolutely voluntary. All participants have received clear explanation of the research aims. Women who agreed to participate were asked to fill out questionnaires at the Clinic, during their visits before oocyte retrieval. Pa-tients could stay alone during the questionnaires filling or with the responsible person if clarifications will be needed (based on their preferences). 

L. 118: You might want to specify that two clinics were in the same city??? 3 clinics - two cities.

Response: The clinics were located in three large cities: Nur-Sultan, Almaty and Shymkent. Eligible participants were recruited from three fertility clinics in Kazakhstan.

There was a typo in the original text. Amendments were done.

L. 125: When did they fill-out the baseine questionnaire?

Response: the questionnaires were filled out during the initial visits to the IVF clinic before the oocytes retrieval.

L. 129: What kind of medical history information was gathered? Previous depressions, anxiety - treatments? Prescribed medication?

Response: The following information was gathered: demographic data, educational level, income level, biomedical data, age, BMI, infertility duration (years) and causes, previous pregnancy history (if any), previous miscarriage, history of other diseases (including psychological), number of oocytes retrieved in previous cycles, number of embryos transferred, number of previous IVF trials, prescribed medications, etc.

The description is present in the text of the methods section. Some data are present in the tables (1-5).

“The baseline questionnaire collected data about socio-demographic characteristics, such as, age, BMI, education level. Also their past medical history information was collected, such as, comor-bidities associated with infertility; infertility duration, which was defined as the time from the date of active child wish, or the date of last miscarriage to the date of the first IVF clinic visit; number of previous deliveries; number of previous miscarriages; number of intentional pregnancy interruptions and number of previous IVF cycles performed. The number of oocytes retrieved, the number of embryos transferred, as well as the cause of infertility (female (tubo-peritoneal, ovarian), male and mixed), the type of treatment protocol, fertilization and implantation rates were documented by a physician”.

L.140-146: Why did you choose CES-D? And what is the average score of the Kazakhstan population? or regional data?

Response: CES-D was chosen for the study as it has become a widely used clinical screening tool for the presence of depression [doi:10.1016/j.jad.2018.02.071].

There is no available publications discussing the average CES-D score of the Kazakhstani population. The population based studies have not been done yet. 

L.187: I believe that The Results section should be a header in itself and not presented under 'methods'.

 Response: The results section is a header by itself. The structure of the manuscript is as it is shown below:

1. Background

2. Methods

3. Results

4. Discussion

5. Conclusion

L. 189: How many did you approach - how many declined to particioate? Do you know why? A flowchart will give us a nice overview.

Response: During the study 2320 patients have been approached in total in three IVF clinics. Out of all approached patients 304 agreed to participate.

Since participation in the study was completely voluntary, we did not feel appropriate to ask those who refused to participate about the nature of the decision. It might be accepted as pressure.

L. 190: definition of normal BMI (also in a cultural perspective). Also it is unclear, if women who conceived were in first, second or more cycles. It would be beneficial to add this information to your table (the frequency of pregnacy rate for each cycle).

Response: There is no specific local/cultural range of BMI in the country. With regards to BMI/obesity, local specialists are guided by international definitions/standards. Normal BMI is considered in the range of 18.5–24.9 kg/m2.

L. 220-222: from what perspective? Do you have any regional data to compare with? Again a table with an overview of the descriptive reginal data would make it much easier to understand the differences.

Response: CES-D overall score was dichotomized using a cut-off 16 and above as at risk for clinical depression. In this study, the Cronbach’s alpha coefficient for CES-D scale was 0.94.

To our knowledge, there is no published available data discussing the average CES-D score of the Kazakhstani population or regional population. The population based studies have not been done yet. This might be a task for the future studies. 

L.230-233: I am a little usure that I understand this paragraph correctly. If you controlled for the variables mentionned then the association between FPI global stress was still sig-nificant or not significant for pregnancy outcome? You may want to consider 'main' anal-yses and then explorative analyses e.g. looking into specific domains association with pregnancy. It is a little muddled.

Response:

Thank you for the comment. Global stress was statistically associated after adjustment for the covariates, mentioned in the text, with the outcome. To make it clearer, we rewrote the whole “3.4 Depression, stress and anxiety” subsection. Please see in the manuscript.

Discussion

Overall, the discussion is long. You may want to shorten it a bit. 

Response: Some sentences have been deleted.

L.251-261: This is a very interesting discussion point indeed, however as you don't your-selves present any biological/physiological data it seems a little odd to present it here. You are speculating whether this might be an explanation for your findings.

Response: The paragraph was rewritten and irrelevant sentences (that are not related to the study results) were deleted.

L.272-281: What is the main reason for this paragraph? You might want to leave it out. As it doesn't add to your main objective.

Response: Thank you for the comment. The paragraph is shortened, however, we would like to keep some sentences as it is linked to the study results.

L. 301: If you did power calculations accordingly to the number of participants you needed, it would be a good idea to present this information in the Methods section, as this is defintely a strength.

Response: 

We appreciate the comment given by the Reviewer. We added a statement about power analysis in the Methods section: “Finally, post hoc power analysis was performed to determine if insufficient sample size have played role in non-significant findings. Power was calculated based on alpha=0.05, a mean difference or the minimally clinically important difference was approximated by one standard error of a measurement (“calculated as the standard deviation of the scale multiplied by the square root of one minus its reliability coefficient”) (REF: Further evidence supporting an SEM-based criterion for identifying meaningful intra-individual changes in health-related quality of life). CES-D scale had the lowest power (32.6%); to obtain statistical power at the recommended 0.80 level sample size for CES-D scale would be needed to be 800. STAI-S (65.6%) and STAI-T (74.9%) had lower than the recommended statistical power level. All FPI subscales had sufficient power (>80%) except Need for parenthood (71.6%).” 

Also, Discussion regarding power was slightly revised and see highlighted revisions.

L. 313-315: This is an interesting result, wy isn't this dicussed further?

Response: In this manuscript we are reporting the study preliminary results. The final manuscript prepared on the study complete data will include expanded discussion.

---

## [Editor Report · Decision Letter 1]

26 Oct 2020

The effect of psychological distress on IVF outcomes: reality or speculations?

PONE-D-20-18832R1

Dear Dr. Aimagambetova,

We’re pleased to inform you that your manuscript has been judged scientifically suitable for publication and will be formally accepted for publication once it meets all outstanding technical requirements.

Kind regards,

Alessio Paffoni, PhD

Academic Editor

PLOS ONE

Additional Editor Comments (optional):

The manuscript met the reviewers' recommendations positively.

The limitations which still exist have been sufficiently discussed. I believe the manuscript meets the requirements for publication at this point.
---

## [Editor Report · Acceptance letter]

4 Dec 2020

PONE-D-20-18832R1 

The effect of psychological distress on IVF outcomes: reality or speculations? 

Dear Dr. Aimagambetova:

I'm pleased to inform you that your manuscript has been deemed suitable for publication in PLOS ONE. Congratulations! Your manuscript is now with our production department. 

Kind regards, 

on behalf of

Dr. Alessio Paffoni 

Academic Editor

PLOS ONE